# Human Dimensions in an Animal Disease Reporting System: A Scoping Review Protocol and Pilot Mapping to Behavioral Frameworks

**Alwyn Tan** [1,2,*]**, Sangeeta Rao** [1] **and Mo Salman** [1]

1    Animal Population Health Institute, College of Veterinary Medicine and Biomedical Sciences, Colorado State University, Campus Stop 1644, Fort Collins, CO 80523, USA; sangeeta.rao@colostate.edu (S.R.); mo.salman@colostate.edu (M.S.)
2    Animal & Veterinary Service, National Parks Board, 1 Cluny Road, Singapore 259569, Singapore
*    Correspondence: alwyn_tan@nparks.gov.sg

**Abstract:** Effective animal disease reporting is critical for early disease detection and control, but it is often hindered by various human behavioral barriers. This review outlines a comprehensive approach to understanding and addressing these barriers in animal owners and producers. The result is a proposed scoping review protocol to find evidence on human behavioral barriers, enablers, and interventions to animal disease reporting and the use of established behavioral frameworks, including the Theoretical Domains Framework (TDF) and Behavior Change Wheel (BCW), to systematically analyze factors affecting disease reporting behavior. This scoping review protocol introduces a novel perspective on animal disease reporting by delving into the human behavioral aspects. By leveraging established frameworks, we aim to provide systematic insights into the influences on animal disease reporting behavior and propose evidence-based interventions. This research has the potential to significantly contribute to the enhancement of global animal health surveillance systems.

**Keywords:** animal health; animal disease; zoonosis; disease surveillance; disease reporting; early detection; behavioral framework; behavioral barriers; behavioral enablers; behavioral interventions

## 1. Introduction

Animal disease reporting by animal owners and producers involves the reporting of clinical or suspect cases of animal disease to the veterinary authorities. In most countries, the veterinary authorities have legal requirements for veterinarians, animal owners, and producers to report suspected cases of notifiable animal diseases to them. It is an important component of passive surveillance in a biosurveillance system to quickly detect cases of animal disease in a population so that actions can be taken to contain and eradicate the disease before it spreads [1,2]. However, animal diseases remain underreported, and commonly cited reasons include the lack of awareness and knowledge of reportable diseases by the stakeholders and the lack of appropriate compensation [3,4].

Even though animal disease reporting by animal producers and owners is an important component of animal health surveillance, there has been limited research on barriers and enablers that influence reporting, and the effectiveness of interventions to improve reporting. A study by Bronner et al. [5] found that veterinarians and farmer associations had a major influence on the reporting of bovine brucellosis by farmers. More in-depth human behavioral analysis could identify specific interventions to implement for the farmer, veterinarian, or associations to improve the participation of the farmers in animal disease reporting. An article by Brugere et al. [6] elaborated on the human dimensions of disease surveillance in aquaculture farms by identifying numerous social, economic, and institutional factors that affected animal disease reporting, in addition to the more commonly explored veterinary, epidemiology, and technical factors. The article recommended that

human dimensions must be considered in the framework for animal disease surveillance but did not include the scope of analyzing and recommending specific behavioral interventions to improve animal disease reporting in aquaculture farmers. The use of behavioral frameworks will allow a systematic way of mapping behavioral barriers identified in the context of aquaculture disease reporting to specific types of interventions. Other articles by Ebata et al. [7], Mariner et al. [8], Struchen et al. [9], and Lupo et al. [10] had similarly advocated the inclusion of human dimensions in the surveillance systems for zoonotic diseases, avian influenza, equine diseases, and oyster diseases, but could be followed up with using behavioral frameworks to find practical interventions in each context.

In a systematic review of social research data collection methods used to investigate animal disease reporting behavior, Enticott et al. [11] recommended more studies on animal disease reporting to include behavioral mechanisms to improve the understanding of how disease reporting works. A scoping review by Gates et al. [12] provided a good overview of factors that influenced animal disease reporting behavior in farmers but did not explain those factors in terms of behavioral theories.

Behavioral theories provide an explanation of the structural and psychological mechanisms believed to control behavior and changes in behavior [13]. Garza et al. [14], Barnes et al. [15], and Fountain et al. [16] utilized behavioral theories such as the nudge theory and Schwartz's theory of basic human values to assess strategies used to increase compliance with animal biosecurity measures and disease reporting to safeguard the health of animals. These studies generally found that the use of behavioral theories was beneficial for increasing the range of interventions that could be considered to effect behavior change and increasing the effectiveness of interventions chosen to improve animal health.

Behavior change frameworks, such as the Theoretical Domains Framework (TDF), are a synthesis of multiple theories and were initially developed to investigate the influences on public health behavior, such as factors that encourage physical activity and interventions to reduce smoking [13,17]. More recently, Michie et al. [18] developed the Behavior Change Wheel (BCW) to help non-behavioral science specialists design interventions for behavior change. The BCW is a synthesis of 19 frameworks of behavior change and recognizes that behavior is influenced by capability, opportunity, and motivation (collectively known as the COM-B model). Behavior change frameworks have also been used to assess behavior change in different sectors such as workplace safety, environmental cleanliness, and consumer food choices [19–21]. In relation to the animal sector, recent reviews have been conducted on and advocate the use of behavior change frameworks to improve the responsible use of antimicrobials on farms, improve animal welfare, and improve disease control measures in cattle farms [22–24]. From these prior studies, we are confident that behavior change frameworks are relevant to studying the factors that lead to the underreporting of animal diseases and interventions to improve reporting.

A scoping review is a method of literature review to synthesize evidence to address a broad research question [25]. We aim to develop a protocol for conducting a scoping review to identify behavioral barriers, enablers, and interventions for animal owners and producers reporting animal diseases to veterinary authorities. The barriers, enablers, and interventions will then be mapped to behavioral frameworks to understand the mechanisms of action that influence disease reporting so that a broader range of interventions guided by theory and evidence can be considered. A demonstration of the proposed protocol, along with its limitations, is also provided.

## 2. Materials and Methods

### 2.1. Scoping Review

Scoping reviews are a rigorous and transparent method of evidence synthesis to address broad and exploratory research questions, such as identifying key characteristics or factors related to a concept [26–28]. Scoping reviews allow for a comprehensive search for evidence as both published and unpublished (i.e., gray literature) primary sources of evi-

dence can be considered [28]. Our objectives were to explore, summarize, and map evidence on animal disease reporting behavior; hence, we chose to conduct a scoping review.

We developed a protocol that aligned with the Preferred Reporting Items for Systematic Reviews and Meta-Analyses extension for Scoping Reviews (PRISMA-ScR) statement [29] which is described in the following subsections.

### 2.2. Research Question and Definitions

We formulated the research question for the scoping review using the PICO (Population, Intervention, Comparator, and Outcome) framework (Table 1), which facilitated the subsequent literature search and screening criteria [28,30]. We defined the question to be addressed by the scoping review as: What are the behavioral barriers, enablers, and interventions for animal owners and producers reporting animal diseases to veterinary authorities?

**Table 1.** Formulation of research question using the PICO (population, intervention, comparator, and outcome) framework.

| Key Elements | Definition | Components of this Study |
|---|---|---|
| Population | Population demographics, characteristics, and other qualifying criteria. | Animal owners and producers. |
| Intervention | Interventions being considered. | Behavioral barriers, enablers, and interventions. |
| Comparator | Comparison of intervention with a control. | Not applicable. |
| Outcomes | Desired effects of the intervention. | Animal disease reporting. |

To ensure a congruent and uniform understanding of the scope of the research question by reviewers and readers, we defined each component in the PICO framework for this study (Table 2).

**Table 2.** Definitions of each component in the research question.

| Components of This Study | Definition |
|---|---|
| Animal owners and producers | All stakeholders that are responsible for the animals under their care. Animals include any terrestrial or aquatic animals, companion animals, livestock, zoo animals, wildlife, and bees. |
| | Examples include pet owners, pet breeders, livestock farmers, livestock smallholders, animal caretakers, zoo animal keepers, veterinarians, game officers, game managers, game wardens, wildlife managers, and hunters. |
| Behavioral barriers and enablers | Factors that negatively or positively influence behavior. |
| | Behavior refers to anything a person (both individual and collective) does in response to internal or external events. Actions may be overt and directly measurable, or covert and indirectly measurable; behaviors are physical events that occur in the body and are controlled by the brain [18,31]. |
| Behavioral interventions | An activity or coordinated set of activities that aims to influence an individual or population to behave differently from how they would have acted without such an action [18]. |

**Table 2.** *Cont.*

| Components of This Study | Definition |
|---|---|
| Animal disease reporting | Animal owners and producers report suspected cases of reportable animal diseases to a veterinary authority. |
| | Reportable animal diseases refer to animal diseases specified by veterinary authorities, and that, as soon as detected or suspected, should be brought to the attention of the authority, in accordance with national regulations [32]. |

### 2.3. Eligibility Criteria

This scoping review included articles on behavioral barriers, enablers, and interventions for animal owners and producers reporting animal diseases to veterinary authorities. The eligibility criteria were developed over multiple iterations of refinement by the reviewers (Table 3).

**Table 3.** Eligibility criteria for the scoping review.

| Characteristics of Sources of Evidence | Revised Eligibility Criteria |
|---|---|
| Types of articles | We included articles published in a peer-reviewed journal or gray literature authored by an international or governmental organization. Commentaries, opinion pieces, and conference abstracts will be excluded as they are unlikely to provide sufficient details or primary information on animal disease reporting behavior required for subsequent mapping to behavior change frameworks. Academic theses and dissertations will be excluded as significant research findings are likely to be published in peer-reviewed journals subsequently. |
| Date range | We included articles published between January 1924 (the year the World Organisation for Animal Health and international standards of animal disease reporting was established) and the date the search was conducted. |
| Language | We only considered English language articles. |
| Population | We included animal owners and producers, with no restriction on geographical regions. |
| Interventions | We included behavioral barriers, enablers, or interventions that are primary information, generated from evidence and not opinions, such as those obtained from interviews, surveys, focus groups, expert elicitations, observational studies, case studies, trials, etc. Behavioral barriers and enablers were included if they were clear, precise, distinct, and observable. Behavioral interventions were included if they were evaluated for the effectiveness of the outcome within their respective studies. |
| Outcome | We included animal disease reporting by animal owners and producers to veterinary authorities. |

### 2.4. Information Sources and Search Strategy

A literature search will be conducted on the following bibliographic databases: AGRICOLA, Aquatic Sciences and Fisheries Abstracts, CAB Abstracts, APA PsychInfo, PubMed, Web of Science, and Zoological Record. Gray literature articles were searched on the World Organisation for Animal Health (WOAH) and World Health Organization (WHO) databases.

Controlled vocabulary such as Medical Subject Headings (MeSH) and truncated search terms were combined using Boolean and proximity operators to reach a balance between sensitivity and specificity when searching for articles. The snowballing strategy was used to

identify additional references from included studies and from published literature reviews (secondary information sources) that are relevant to the scope of this study.

An example of a draft search strategy for PubMed can be found in Appendix A. Similar search terms with modifications to the syntax were used for the other databases. The search results were exported to EndNote 20 (Version 20.6, Clarivate Plc, London, UK), and duplicates were removed. The deduplicated search results were then exported to the DistillerSR program (Version 2023.1, DistillerSR Inc., Ottawa, ON, Canada) for screening, critical appraisal, and data charting. It was expected that the use of a systematic review software with more integrated and machine learning functions would increase the speed and reduce errors in conducting the scoping review.

### 2.5. Selection of Sources of Evidence

Two independent reviewers evaluated the titles and abstracts of articles identified in the searches to exclude articles that were not relevant to this study. Disagreements were resolved through discussion and consensus. Following this, two independent reviewers evaluated the full text of the articles to confirm the inclusion of articles that passed the title and abstract screening. Similarly, disagreements at this step were resolved through discussion and consensus. The reviewers explored the use and availability of artificial intelligence algorithms to facilitate or automate the literature screening process. The selection process was recorded and the PRISMA flow diagram is provided below [29].

### 2.6. Data Charting Process and Data Items

Data charting was conducted independently by two reviewers on a form created in DistillerSR. The results were discussed and continuously updated in an iterative process. The data extraction form included the following variables: authors, year of publication, peer-reviewed or gray literature, country of study, study objectives, animal disease(s) specified, whether disease(s) specified were zoonotic, behavioral barriers, behavioral enablers, and behavioral interventions.

### 2.7. Critical Appraisal of Sources of Evidence

The quality of the articles was appraised using criteria adapted from the Mixed Methods Appraisal Tool (MMAT) Version 2018 [33]. Articles were appraised on whether they had clear research questions and whether the data collected addressed the research questions in the respective articles. Two reviewers independently appraised the included articles. Disagreements were resolved through discussion and consensus.

### 2.8. Data Analysis and Synthesis of Results

Descriptive statistics were used to describe the type of literature included, countries of study, the context of animal diseases, and whether they were zoonotic.

Behavioral barriers and enablers to animal disease reporting were mapped to domains of the Theoretical Domains Framework (TDF) [34] and components of the COM-B model [18].

Behavioral interventions to animal disease reporting were mapped to Behavior Change Techniques (BCT) of the Behavior Change Techniques Taxonomy version 1 (BCTTv1) [35] and intervention functions of the BCW [18].

We analyzed the behavioral barriers, enablers, and interventions that have been coded to the respective behavioral frameworks and presented our findings on animal disease reporting from a socio-behavioral perspective. We also discuss the implications of animal health surveillance and suggest potential strategies to improve animal disease reporting.

### 3. Results and Demonstrations

A pilot scoping review was conducted to demonstrate the mapping of behavioral barriers to behavioral frameworks and interventions. In the pilot, a search for relevant articles was conducted on the Web of Science and Scopus databases.

The search strategy used in the pilot study is detailed in Appendix B. The search results were exported to EndNote 20, and duplicates were removed. The deduplicated search results were then exported to Rayyan (Rayyan Systems Inc., Cambridge, MA, USA) for screening. Critical appraisal and data charting were performed in forms on Microsoft Excel (Version 2312, Microsoft Corp., Redmond, WA, USA).

The pilot search extracted 125 articles. A random sample of 25 articles was selected using random numbers generated in Microsoft Excel. Three independent reviewers screened the 25 articles. Thirteen articles were screened in full text, and seven articles were excluded because they did not fulfill the eligibility criteria (Figure 1). All six included articles [3,5,36–39] were peer-reviewed journal articles and were appraised to have clear research questions with data collected to address their respective research questions.

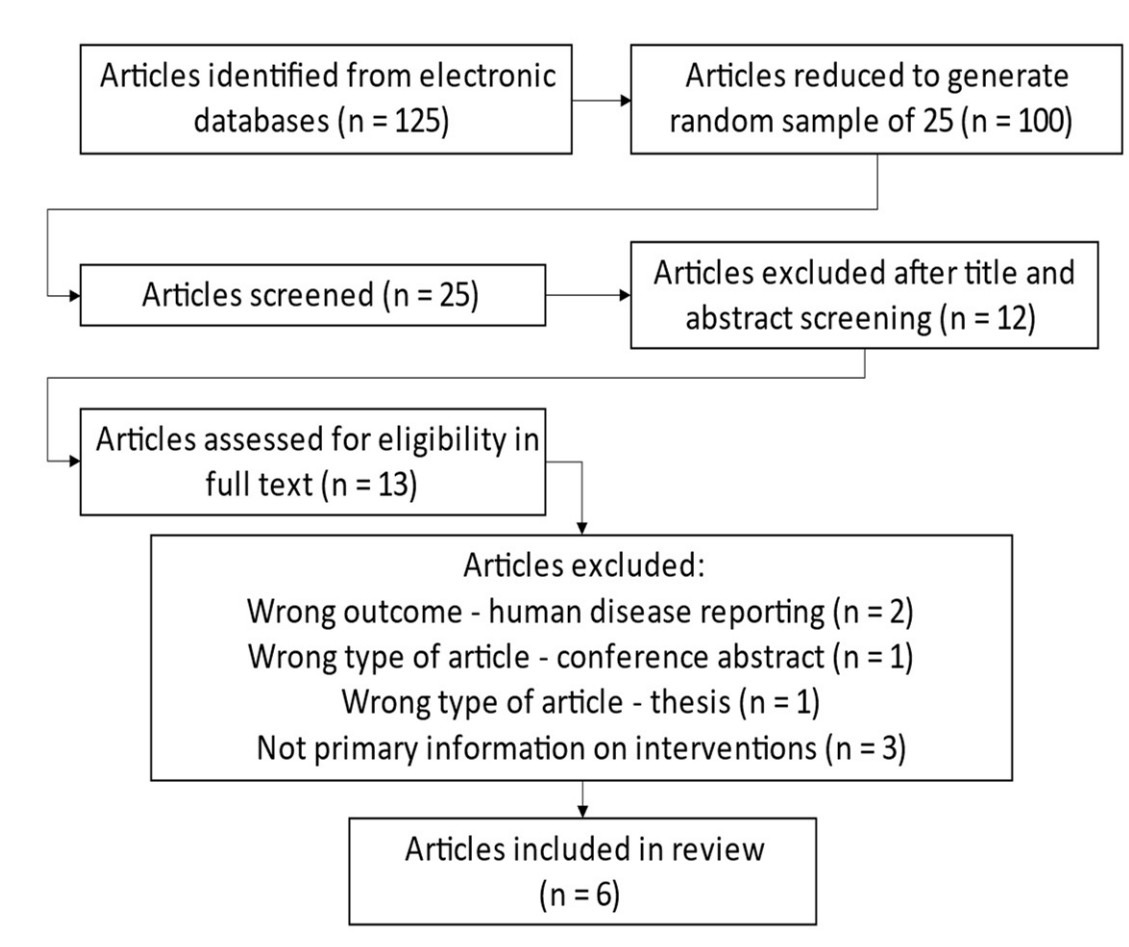

**Figure 1.** Flow diagram from bibliographic search of articles to final inclusion in the pilot review.

Details of the included articles are summarized in Appendix C. The articles covered 11 countries distributed across two continents (Europe and Oceania). In terms of the population of disease reporting, five articles covered animal producers, one covered animal hunters, and one covered veterinarians. In terms of animal disease, four articles covered specific animal diseases, and of these, two were zoonotic diseases. A total of five articles contained primary information on behavioral barriers, five articles contained primary information on behavioral enablers, and no articles contained primary information on behavioral interventions.

We present examples to demonstrate the mapping of evidence on behavioral barriers extracted from two articles using both the TDF (Table 4) and COM-B models (Table 5), linking them to BCT and BCW intervention functions that are likely to be effective in increasing disease reporting behavior, respectively. These links were identified through a consensus exercise among groups of experts and should be thought of as options that can be applied as interventions. For example, Elbers et al. [3] found that one barrier to disease reporting was that "farmers felt that during past animal disease eradication campaigns they were pushed aside and they were not in control of their business anymore". This negative interpersonal experience affected the social opportunity (COM-B component) for disease reporting. Based on this, one possible environmental restructuring (intervention function) is that during eradication campaigns, the veterinary authorities designate relationship managers to farmers to receive their feedback and to provide updates on the operations.

**Table 4.** Mapping text on behavioral barriers to TDF and linking them to BCT.

| Article | Text Description of Barrier | TDF Domain | BCTs Likely to Be Effective |
|---|---|---|---|
| Elbers et al. [3] | Farmers felt that during past animal disease eradication campaigns, they were pushed aside and they were not in control of their business anymore. | Social influences | • Social comparison<br>• Social support or encouragement (general)<br>• Information about others' approval<br>• Social support (emotional)<br>• Social support (practical)<br>• Vicarious reinforcement<br>• Restructuring the social environment<br>• Modelling or demonstrating the behavior<br>• Identification of self as a role model<br>• Social reward |
| Elbers et al. [3] | In the layer sector in the Netherlands, there is almost no regular veterinary supervision, and health problems are commonly discussed with technical/non-veterinary advisers from poultry integrations or the feed industry. | Environmental context and resources | • Restructuring the physical environment<br>• Discriminate (learned) cue<br>• Prompts/cues<br>• Restructuring the social environment<br>• Avoidance/changing exposure to cues for the behavior |
| Vergne et al. [36] | Hunters indicating that they do not report the presence of wild boar carcasses frequently attributed this behavior to being unaware of the possibility to report. | Knowledge | • Health consequences<br>• Biofeedback<br>• Antecedents<br>• Feedback on behavior |

**Table 5.** Mapping text on behavioral barriers to the COM-B model and linking them to BCW intervention functions.

| Article | Text Description of Barrier | COM-B Component | BCW Intervention Functions Likely to Be Effective |
|---|---|---|---|
| Elbers et al. [3] | Farmers felt that during past animal disease eradication campaigns, they were pushed aside and they were not in control of their business anymore. | Social opportunity | • Restriction<br>• Environmental restructuring<br>• Modelling<br>• Enablement |
| Elbers et al. [3] | In the layer sector in the Netherlands, there is almost no regular veterinary supervision, and health problems are commonly discussed with technical/non-veterinary advisers from poultry integrations or the feed industry. | Physical opportunity | • Training<br>• Restriction<br>• Environmental restructuring<br>• Enablement |
| Vergne et al. [36] | Hunters indicating that they do not report the presence of wild boar carcasses frequently attributed this behavior to being unaware of the possibility to report. | Psychological capability | • Education<br>• Training<br>• Environmental restructuring<br>• Modelling<br>• Enablement |

It should be noted that although these suggested BCTs and BCW intervention functions do not provide specific designs of behavior change interventions, they do provide a systematic and theoretically guided method for identifying the types of interventions that are expected to be effective for increasing animal disease reporting (behavior) by animal owners and producers (target population) [18]. Specific intervention design will ultimately depend on the local context and circumstances [18,40].

## 4. Discussion

### 4.1. Using Behavioral Frameworks to Understand Behavior and Behavior Change

The action of animal disease reporting by animal owners and producers is a desired outcome in this study. It is difficult to change behavior, such as increasing the rate of animal disease reporting, but the use of behavior change theories and evidence can increase the effectiveness of behavioral interventions [34]. To effectively implement behavior change, we should understand the factors that influence the desired behavior and perform interventions that can positively alter those factors. Behavioral theories are used to understand the mechanisms behind barriers and enablers to disease reporting and guide the selection and implementation of interventions to change behavior.

There are many behavioral theories with overlapping ideas which make it challenging for practical use by implementers [13]. Hence, Michie et al. [41] and Cane et al. [34] developed and refined the Theoretical Domains Framework (TDF), which is a synthesis of 33 behavior and behavior change theories whose constructs are integrated and clustered into 14 domains that cover the cognitive, affective, social and environmental factors that influence behavior. A simplified framework, the COM-B model, has six components (physical capability, psychological capability, physical opportunity, social opportunity, reflective motivation, and autonomic motivation) that influence behavior, and each of these components is linked to the TDF [18]. The use of the simpler COM-B model increases the accessibility of behavioral frameworks to researchers who may not be experts in social and behavioral sciences but may not allow granularity to differentiate influences on a particular behavior. For example, the COM-B model may not provide sufficient details on the influences of psychological capability and reflective motivation compared to the TDF [13]. Nevertheless, when either the TDF or COM-B model is used for behavioral

analysis, the domains or components that are identified to have a large influence on the desired behavior can be used to inform the types of interventions to change that behavior [18,34].

The behavior change technique taxonomy (BCTT) is a standardized language used in describing, analyzing, and implementing interventions, thus avoiding uncertainty and confusion when different non-standardized labels are used. Behavior change interventions have different mechanisms of effecting change, known as behavior change techniques (BCT). A BCT is an observable, replicable, and irreducible component of a behavioral intervention to change causal processes that influence behavior [35]. The BCTT is a standardized labeling of 93 distinct BCTs in 16 clusters developed by Michie et al. [35] as the 'active ingredients' of behavior change interventions. The benefits of using the BCTT include allowing the accurate implementation of effective interventions, the accurate replication of interventions for comparative research, the reliable extraction of information on interventions for reviews, and a better understanding of mechanisms of action and intervention development [35]. A simplified framework of BCTs, comprising nine intervention functions in the Behavior Change Wheel (BCW), facilitates non-behavioral science specialists in designing interventions to change behavior [18].

In summary, the TDF and COM-B models are used to identify what factors need to change to achieve the desired behavior. The BCT and BCW intervention functions are used to determine potential strategies to improve animal disease reporting. The identification of TDF domains or COM-B components influencing animal disease reporting will inform the types of BCT or BCW intervention functions that will likely be effective in bringing about the desired behavior change [18].

### 4.2. Limitations of Study

First, the exclusion of articles in languages other than English may omit potential sources of evidence on animal disease reporting. Second, there may be difficulty mapping text on behavioral barriers, enablers, and interventions to labels of behavioral frameworks as articles may not provide sufficiently detailed descriptions. This is partially overcome by having two independent reviewers conducting the mapping. If there remains uncertainty on the mapping of text and if consensus cannot be reached, the text is proposed to be excluded from mapping.

### 5. Conclusions

This scoping review protocol is a novel effort to investigate and address the human behavioral dimensions of animal disease reporting by animal owners and producers. Based on the presented scoping review protocol, the results of the scoping review performed in the future will allow us to identify specific interventions, guided by BCTs and the BCW to enhance animal disease reporting by animal owners and producers. This research endeavors to bridge the gap between human behavioral science and animal disease reporting, shedding light on the complex human behavioral factors influencing reporting behavior. Our ultimate goal is to empower a more robust and efficient animal health surveillance system, thus safeguarding both animal and public health.

**Author Contributions:** Conceptualization, A.T., S.R. and M.S.; writing—original draft preparation, A.T.; writing—review and editing, A.T., S.R. and M.S.; visualization, A.T. All authors have read and agreed to the published version of the manuscript.

**Funding:** This research received no external funding.

**Institutional Review Board Statement:** Not applicable.

**Data Availability Statement:** The data presented in this study are available in the article.

**Acknowledgments:** The authors thank Stephanie Ritchie, Elizabeth Tobey, and Jessica Sigman from the United States Department of Agriculture (USDA) National Agricultural Library for advising on the protocols for the scoping review and developing the search strategy for the scoping review. We also thank Mary Foley from the USDA Animal & Plant Health Inspection Service (APHIS), Wildlife Services for conducting the literature search that was used in the pilot. Lastly, we thank Deb Green for being a reviewer in the pilot.

**Conflicts of Interest:** The authors declare no conflicts of interest.

## Appendix A

Draft search strategy for PubMed database in the full study.

**Table A1.** Draft PubMed search strategy, including the use of controlled vocabulary, Medical Subject Headings (MeSH), and proximity operators.

| Key Elements | Components of This Study | Search Terms |
|---|---|---|
| Population | Animal owners and producers | farmer*[Title/Abstract] OR producer*[Title/Abstract] OR "farm manager*"[Title/Abstract] OR " farm operator*"[Title/Abstract] OR "livestock manager*"[Title/Abstract] OR "agricultural worker*"[Title/Abstract] OR ("animal owner"[Title/Abstract: ~3] OR "animal owners"[Title/Abstract:~3] OR "animals owner"[Title/Abstract:~3] OR "animals owners"[Title/Abstract:~3]) OR ("pet owner"[Title/Abstract:~3] OR "pet owners"[Title/Abstract:~3] OR "pets owner"[Title/Abstract:~3] OR "pets owners"[Title/Abstract:~3]) OR "pet parent*"[Title/Abstract] OR breeder*[Title/Abstract] OR "owned animal*"[Title/Abstract] OR ("animal keeper"[Title/Abstract:~3] OR "animal keepers"[Title/Abstract:~3] OR "animals keeper"[Title/Abstract:~3] OR "animals keepers"[Title/Abstract:~3]) OR "zoo keeper*"[Title/Abstract] OR zookeeper*[Title/Abstract] OR ("livestock keeper"[Title/Abstract:~3] OR "livestock keepers"[Title/Abstract:~3]) OR veterinarian*[Title/Abstract] OR "vet tech*"[Title/Abstract] OR "vet technician*"[Title/Abstract] OR ("animal caretaker"[Title/Abstract:~3] OR "animal caretakers"[Title/Abstract:~3] OR "animals caretaker"[Title/Abstract:~3] OR "animals caretakers"[Title/Abstract:~3]) OR "animal care staff"[Title/Abstract] OR hobbyist*[Title/Abstract] OR "livestock smallholder*"[Title/Abstract] OR "animal smallholder*"[Title/Abstract] OR ("animal trainer"[Title/Abstract:~3] OR "animal trainers"[Title/Abstract:~3] OR "animals trainer"[Title/Abstract:~3] OR "animals trainers"[Title/Abstract:~3]) OR ("livestock trainer"[Title/Abstract:~3] OR "livestock trainers"[Title/Abstract:~3]) OR jockey*[Title/Abstract] OR "track management"[Title/Abstract] OR grooms[Title/Abstract] OR "racing association"[Title/Abstract] OR "racing authorit*"[Title/Abstract] OR habits[Title/Abstract] OR cowhand*[Title/Abstract] OR "incentives"[Title/Abstract] OR cowboy*[Title/Abstract] OR rancher*[Title/Abstract] OR "ranch worker*"[Title/Abstract] OR herdsmen[Title/Abstract] OR herder*[Title/Abstract] OR shepherd*[Title/Abstract] OR aquaculturist*[Title/Abstract] OR beekeeper*[Title/Abstract] OR apiculturist*[Title/Abstract] OR biologist*[Title/Abstract] OR "environmental scientist*"[Title/Abstract] OR gamekeeper*[Title/Abstract] OR "game warden*"[Title/Abstract] OR "game keeper*"[Title/Abstract] OR ranger*[Title/Abstract] OR "rehabilitator"[Title/Abstract] OR "animal shelter*"[Title/Abstract] OR zoologist*[Title/Abstract] OR hunter*[Title/Abstract] OR "control operator*"[Title/Abstract] OR officer*[Title/Abstract] OR "Farmers"[Mesh] OR "Veterinarians"[Mesh] OR "Animal Technicians"[Mesh] OR "Laboratory Personnel"[Mesh] |
| Intervention | Behavioral barriers, enablers, and interventions | psycholog*[Title/Abstract] OR "acceptance"[Title/Abstract] OR "access"[Title/Abstract] OR "accountability"[Title/Abstract] OR adaptability[Title/Abstract] OR "adopt*"[Title/Abstract] OR "administrate"[Title/Abstract] OR "administrative"[Title/Abstract] OR "altruism"[Title/Abstract] OR "assistance"[Title/Abstract] OR attitude*[Title/Abstract] OR "avoidance"[Title/Abstract] OR barrier*[Title/Abstract] OR "approach Behavior"[Title/Abstract] OR "planned behavior"[Title/Abstract] OR "reporting behavior*"[Title/Abstract] OR "approach Behaviour"[Title/Abstract] OR "planned behaviour"[Title/Abstract] OR "reporting behaviour*"[Title/Abstract] OR ("behavior choice"[Title/Abstract:~3] OR "behavior choices"[Title/Abstract:~3] OR "discrete choice"[Title/Abstract:~3] OR "discrete choices"[Title/Abstract:~3] OR "behaviour choice"[Title/Abstract:~3] OR "behaviour choices"[Title/Abstract:~3]) OR belief*[Title/Abstract] OR "blame"[Title/Abstract] OR budget*[Title/Abstract] OR "capacity"[Title/Abstract] OR "choice"[Title/Abstract] OR "cognitive"[Title/Abstract] OR "collaborat*"[Title/Abstract] OR "complexity"[Title/Abstract] OR "compliance"[Title/Abstract] OR "cooperation"[Title/Abstract] OR "co-operation"[Title/Abstract] OR "cost benefit"[Title/Abstract] OR "credibility"[Title/Abstract] OR "cues"[Title/Abstract] OR "curiosity"[Title/Abstract] OR ("decision report"[Title/Abstract:~3] OR "decisions report"[Title/Abstract:~3] OR "decision reporting"[Title/Abstract:~3] OR "decisions reporting"[Title/Abstract:~3]) OR "make decision"[Title/Abstract:~3] OR "make decisions"[Title/Abstract:~3] OR "making decision"[Title/Abstract:~3] OR "making decisions"[Title/Abstract:~3] OR "made decision"[Title/Abstract:~3] OR "made decisions"[Title/Abstract:~3] OR "maker decision"[Title/Abstract:~3] OR "maker decisions"[Title/Abstract:~3] OR "decided"[Title/Abstract] OR "economics"[Title/Abstract] OR "emotional"[Title/Abstract] OR "emotions"[Title/Abstract] OR facilitat*[Title/Abstract] OR fear*[Title/Abstract] OR "framing effect*"[Title/Abstract] OR habit[Title/Abstract] OR "help-seeking"[Title/Abstract] OR "incentives"[Title/Abstract] OR "incentivize"[Title/Abstract] OR indeterminac*[Title/Abstract] OR "inexperienced"[Title/Abstract] OR "information seeking"[Title/Abstract] OR instinct*[Title/Abstract] OR intention*[Title/Abstract] OR "intervention*"[Title/Abstract] OR "involvement"[Title/Abstract] OR "Knowledge, attitudes and practices"[Title/Abstract] OR "knowledge attitude* practice*"[Title/Abstract] OR "KAP"[Title/Abstract] OR "liability"[Title/Abstract] OR motivat*[Title/Abstract] OR neuroecon*[Title/Abstract] OR "obedience"[Title/Abstract] OR prejudice*[Title/Abstract] OR perception*[Title/Abstract] OR "practices"[Title/Abstract] OR "prompting"[Title/Abstract] OR "priming"[Title/Abstract] OR "prosocial behavior"[Title/Abstract] OR "prosocial behaviour"[Title/Abstract] OR psychosocial*[Title/Abstract] OR "public awareness"[Title/Abstract] OR resilience[Title/Abstract] OR "resistance"[Title/Abstract] OR rationale*[Title/Abstract] OR "reasoning"[Title/Abstract] OR "recognise"[Title/Abstract] OR "recognize"[Title/Abstract] OR "reflexes"[Title/Abstract] OR "reputation"[Title/Abstract] OR respond*[Title/Abstract] OR reponse*[Title/Abstract] OR "responsibility"[Title/Abstract] OR "risk perception"[Title/Abstract] OR ("risks perception"[Title/Abstract:~3] OR "risk perceptions"[Title/Abstract:~3] OR "risks perceptions"[Title/Abstract:~3] OR "risk perceive"[Title/Abstract:~3] OR "risks perceive"[Title/Abstract:~3] OR "risk perceives"[Title/Abstract:~3] OR "risks perceives"[Title/Abstract:~3] OR "risk perceived"[Title/Abstract:~3] OR "risks perceived"[Title/Abstract:~3] OR "risk perceiving"[Title/Abstract:~3] OR "risks perceiving"[Title/Abstract:~3]) OR "self-efficacy"[Title/Abstract] OR sensibilit*[Title/Abstract] OR "shame"[Title/Abstract] OR "shaming"[Title/Abstract] OR "sharing"[Title/Abstract] OR "social network"[Title/Abstract] OR strateg*[Title/Abstract] OR stress[Title/Abstract] OR "surveillance network*"[Title/Abstract] OR "suspicion"[Title/Abstract] OR "tolerance"[Title/Abstract] OR "trust"[Title/Abstract] OR uncertain*[Title/Abstract] OR "under-reporting"[Title/Abstract] OR "underreporting"[Title/Abstract] OR "values"[Title/Abstract] OR "veterinary herd health management"[Title/Abstract] OR "vigilan*"[Title/Abstract] OR "willingness"[Title/Abstract] OR "Psychology"[Mesh] OR "Social Responsibility"[Mesh] OR "Organization and Administration"[Mesh] OR "Altruism"[Mesh] OR "Social Behavior"[Mesh] OR "Helping Behavior"[Mesh] OR "Attitude"[Mesh] OR "Information Avoidance"[Mesh] OR "Communication Barriers"[Mesh] OR "Behavior"[Mesh] OR "Health Belief Model"[Mesh] OR "Budgets"[Mesh] OR "Choice Behavior"[Mesh] OR "Cognitive Psychology"[Mesh] OR "Cognitive Neuroscience"[Mesh] OR "Intersectoral Collaboration"[Mesh] OR "Community Participation"[Mesh] OR "International Cooperation"[Mesh] OR "Cost-Benefit Analysis"[Mesh] OR "Cues"[Mesh] OR "Exploratory Behavior"[Mesh] OR "Decision Making"[Mesh] OR "Emotions"[Mesh] OR "Fear"[Mesh] OR "Habits"[Mesh] OR "Help-Seeking Behavior"[Mesh] OR "Instinct"[Mesh] OR "Intention"[Mesh] OR "Psychosocial Intervention"[Mesh] OR "Early Medical Intervention"[Mesh] OR "Health Knowledge, Attitudes, Practice"[Mesh] OR "Liability, Legal"[Mesh] OR "Motivation"[Mesh] OR "Prejudice"[Mesh] OR "Perception"[Mesh] OR "Psychosocial Functioning"[Mesh] OR "Resilience, Psychological"[Mesh] OR "Public Opinion"[Mesh] OR "Recognition, Psychology"[Mesh] OR "Reflex"[Mesh] OR "Risk"[Mesh] OR "Health Risk Behaviors"[Mesh] OR "Risk Reduction Behavior"[Mesh] OR "Risk Management"[Mesh] OR "Risk Sharing, Financial"[Mesh] OR "Risk Assessment"[Mesh] OR "Risk Factors"[Mesh] OR "Risk Adjustment"[Mesh] OR "Risk-Taking"[Mesh] OR "Risk Evaluation and Mitigation"[Mesh] OR "Social Perception"[Mesh] OR "Self Efficacy"[Mesh] OR "Shame"[Mesh] OR "Cost Sharing"[Mesh] OR "Social Network Analysis"[Mesh] OR "Social Networking"[Mesh] OR "Adaptation, Psychological"[Mesh] OR "Stress, Psychological"[Mesh] OR "Social Stigma"[Mesh] OR "Community Networks"[Mesh] OR "Trust"[Mesh] OR "Uncertainty"[Mesh] OR "Social Values"[Mesh] OR "Social Norms"[Mesh] |
| Outcomes 1 | Disease reporting | "disease tracking"[Title/Abstract] OR "disease tracker"[Title/Abstract] OR "disease tracked"[Title/Abstract:~3] OR "disease track"[Title/Abstract:~3] OR "diseases tracking"[Title/Abstract:~3] OR "diseases tracker"[Title/Abstract:~3] OR "disease track"[Title/Abstract:~3] OR "diseases track"[Title/Abstract:~3] OR "diseases tracks"[Title/Abstract:~3] OR "diseased tracking"[Title/Abstract:~3] OR "diseased tracker"[Title/Abstract:~3] OR "diseased tracked"[Title/Abstract:~3] OR "diseased track"[Title/Abstract:~3] OR "diseased tracks"[Title/Abstract:~3] OR "disease survey"[Title/Abstract:~3] OR "disease surveys"[Title/Abstract:~3] OR "disease surveying"[Title/Abstract:~3] OR "disease surveyer"[Title/Abstract:~3] OR "disease surveyed"[Title/Abstract:~3] OR "diseases surveying"[Title/Abstract:~3] OR "diseases surveyer"[Title/Abstract:~3] OR "diseases surveyed"[Title/Abstract:~3] OR "diseases survey"[Title/Abstract:~3] OR "diseases surveys"[Title/Abstract:~3] OR "diseased surveying"[Title/Abstract:~3] OR "diseased surveyer"[Title/Abstract:~3] OR "diseased surveyed"[Title/Abstract:~3] OR "disease survey"[Title/Abstract:~3] OR "diseased surveys"[Title/Abstract:~3] OR "disease questionnaire"[Title/Abstract:~3] OR "diseases questionnaire"[Title/Abstract:~3] OR "diseased questionnaire"[Title/Abstract:~3] OR "disease surveillance"[Title/Abstract:~3] OR "diseases surveillance"[Title/Abstract:~3] OR "diseased surveillance"[Title/Abstract:~3] OR "disease identifing"[Title/Abstract:~3] OR "disease identifier"[Title/Abstract:~3] OR "disease identifed"[Title/Abstract:~3] OR "disease identify"[Title/Abstract:~3] OR "diseases identifes"[Title/Abstract:~3] OR "diseases identifing"[Title/Abstract:~3] OR "diseases identifier"[Title/Abstract:~3] OR "diseases identifed"[Title/Abstract:~3] OR "diseases identify"[Title/Abstract:~3] OR "diseases identifies"[Title/Abstract:~3] OR "diseased identifing"[Title/Abstract:~3] OR "diseased identifier"[Title/Abstract:~3] OR "diseased identifed"[Title/Abstract:~3] OR "diseased identify"[Title/Abstract:~3] OR "diseased identifies"[Title/Abstract:~3] OR "disease reporting"[Title/Abstract:~3] OR "disease reporter"[Title/Abstract:~3] OR "disease reported"[Title/Abstract:~3] OR "disease report"[Title/Abstract:~3] OR "disease reports"[Title/Abstract:~3] OR "diseases reporting"[Title/Abstract:~3] OR "diseases reporter"[Title/Abstract:~3] OR "diseases reported"[Title/Abstract:~3] OR "diseases report"[Title/Abstract:~3] OR "diseases reports"[Title/Abstract:~3] OR "diseased reporting"[Title/Abstract:~3] OR "diseased reporter"[Title/Abstract:~3] OR "diseased reported"[Title/Abstract:~3] OR "diseased report"[Title/Abstract:~3] OR "diseased reports"[Title/Abstract:~3] OR "disease detecting"[Title/Abstract:~3] OR "disease detection"[Title/Abstract:~3] OR "disease detected"[Title/Abstract:~3] OR "disease detect"[Title/Abstract:~3] OR "disease detects"[Title/Abstract:~3] OR "diseases detecting"[Title/Abstract:~3] OR "diseases detection"[Title/Abstract:~3] OR "diseases detected"[Title/Abstract:~3] OR "diseases detect"[Title/Abstract:~3] OR "diseases detects"[Title/Abstract:~3] OR "diseased detecting"[Title/Abstract:~3] OR "diseased detection"[Title/Abstract:~3] OR "diseased detect"[Title/Abstract:~3] OR "diseased detects"[Title/Abstract:~3] OR "disease controlling"[Title/Abstract:~3] OR "disease controlled"[Title/Abstract:~3] OR "disease control"[Title/Abstract:~3] OR "disease controls"[Title/Abstract:~3] OR "diseases controlling"[Title/Abstract:~3] OR "diseases controlled"[Title/Abstract:~3] OR "diseases control"[Title/Abstract:~3] OR "diseases controls"[Title/Abstract:~3] OR "diseased controlling"[Title/Abstract:~3] OR "diseased controlled"[Title/Abstract:~3] OR "diseased control"[Title/Abstract:~3] OR "diseased controls"[Title/Abstract:~3] OR "disease data collecting"[Title/Abstract:~3] OR "disease data collection"[Title/Abstract:~3] OR "disease data collected"[Title/Abstract:~3] OR "disease data collect"[Title/Abstract:~3] OR "disease data collects"[Title/Abstract:~3] OR "diseases data collecting"[Title/Abstract:~3] OR "diseases data collection"[Title/Abstract:~3] OR "diseases data collected"[Title/Abstract:~3] OR "diseases data collect"[Title/Abstract:~3] OR "diseases data collects"[Title/Abstract:~3] OR "diseased data collecting"[Title/Abstract:~3] OR "diseased data collection"[Title/Abstract:~3] OR "diseased data collected"[Title/Abstract:~3] OR "diseased data collect"[Title/Abstract:~3] OR "diseased data collects"[Title/Abstract:~3] OR "disease census"[Title/Abstract:~3] OR "disease censused"[Title/Abstract:~3] OR "disease censusing"[Title/Abstract:~3] OR "disease censuses"[Title/Abstract:~3] OR "diseases censusing"[Title/Abstract:~3] OR "diseases censused"[Title/Abstract:~3] OR "diseases census"[Title/Abstract:~3] OR "diseases censuses"[Title/Abstract:~3] OR "diseased censusing"[Title/Abstract:~3] OR "diseased census"[Title/Abstract:~3] OR "diseased censused"[Title/Abstract:~3] OR "diseased censuses"[Title/Abstract:~3] OR "disease monitor"[Title/Abstract:~3] OR "disease monitored"[Title/Abstract:~3] OR "disease monitoring"[Title/Abstract:~3] OR "disease monitors"[Title/Abstract:~3] OR "diseases monitoring"[Title/Abstract:~3] OR "disease monitored"[Title/Abstract:~3] OR "diseases monitor"[Title/Abstract:~3] OR "diseases monitors"[Title/Abstract:~3] OR "diseased monitoring"[Title/Abstract:~3] OR "diseased monitors"[Title/Abstract:~3] OR "diseased monitor"[Title/Abstract:~3] OR "diseased monitored"[Title/Abstract:~3] OR "disease communicate"[Title/Abstract:~3] OR "disease communicated"[Title/Abstract:~3] OR "disease communication"[Title/Abstract:~3] OR "disease communicates"[Title/Abstract:~3] OR "disease communicating"[Title/Abstract:~3] OR "disease communications"[Title/Abstract:~3] OR "diseases communicate"[Title/Abstract:~3] OR "diseases communicated"[Title/Abstract:~3] OR "diseases communication"[Title/Abstract:~3] OR "diseases communicates"[Title/Abstract:~3] OR "diseases communicating"[Title/Abstract:~3] OR "diseases communications"[Title/Abstract:~3] OR "diseased communicate"[Title/Abstract:~3] OR "diseased communicated"[Title/Abstract:~3] OR "diseased communication"[Title/Abstract:~3] OR "diseased communicates"[Title/Abstract:~3] OR "diseased communicating"[Title/Abstract:~3] OR "diseased communications"[Title/Abstract:~3] OR "disease record"[Title/Abstract:~3] OR "disease recording"[Title/Abstract:~3] OR "disease records"[Title/Abstract:~3] OR "disease recorded"[Title/Abstract:~3] OR "diseases record"[Title/Abstract:~3] OR "diseases recorded"[Title/Abstract:~3] OR "diseases records"[Title/Abstract:~3] OR "diseases recording"[Title/Abstract:~3] OR "diseased record"[Title/Abstract:~3] OR "diseased recorded"[Title/Abstract:~3] OR "diseased records"[Title/Abstract:~3] OR "diseased recording"[Title/Abstract:~3] OR "disease sampling"[Title/Abstract:~3] OR "disease sampled"[Title/Abstract:~3] OR "diseases sampled"[Title/Abstract:~3] OR "diseases sampling"[Title/Abstract:~3] OR "diseased sampled"[Title/Abstract:~3] OR "diseased sampling"[Title/Abstract:~3] OR "disease exchange"[Title/Abstract:~3] OR "disease exchanging"[Title/Abstract:~3] OR "disease exchanges"[Title/Abstract:~3] OR "disease exchanged"[Title/Abstract:~3] OR "diseases exchange"[Title/Abstract:~3] OR "diseases exchanged"[Title/Abstract:~3] OR "diseases exchanging"[Title/Abstract:~3] OR "diseased exchange"[Title/Abstract:~3] OR "diseased exchanged"[Title/Abstract:~3] OR "diseased exchanges"[Title/Abstract:~3] OR "diseased exchanging"[Title/Abstract:~3] OR "disease disclose"[Title/Abstract:~3] OR "disease disclosing"[Title/Abstract:~3] OR "disease discloses"[Title/Abstract:~3] OR "disease disclosed"[Title/Abstract:~3] OR "diseases disclose"[Title/Abstract:~3] OR "diseases disclosed"[Title/Abstract:~3] OR "diseases disclosing"[Title/Abstract:~3] OR "diseases disclose"[Title/Abstract:~3] OR "diseased disclosing"[Title/Abstract:~3] OR "diseased disclose"[Title/Abstract:~3] OR "diseased disclosed"[Title/Abstract:~3] OR "disease notify"[Title/Abstract:~3] OR "disease notification"[Title/Abstract:~3] OR "disease notifications"[Title/Abstract:~3] OR "disease notified"[Title/Abstract:~3] OR "disease notifies"[Title/Abstract:~3] OR "diseases notify"[Title/Abstract:~3] OR "diseases notifying"[Title/Abstract:~3] OR "diseases notification"[Title/Abstract:~3] OR "diseases notifications"[Title/Abstract:~3] OR "diseases notified"[Title/Abstract:~3] OR "diseases notifies"[Title/Abstract:~3] OR "diseased notifying"[Title/Abstract:~3] OR "diseased notification"[Title/Abstract:~3] OR "diseased notifications"[Title/Abstract:~3] OR "diseased notified"[Title/Abstract:~3] OR "diseased notifies"[Title/Abstract:~3] OR "disease mapping"[Title/Abstract:~3] OR "disease mapped"[Title/Abstract:~3] OR "diseases mapped"[Title/Abstract:~3] OR "diseases mapping"[Title/Abstract:~3] OR "diseased mapped"[Title/Abstract:~3] OR "diseased mapping"[Title/Abstract:~3] OR "disease screening"[Title/Abstract:~3] OR "disease screened"[Title/Abstract:~3] OR "diseases screening"[Title/Abstract:~3] OR "diseases screened"[Title/Abstract:~3] OR "diseased screening"[Title/Abstract:~3] OR "diseased screened"[Title/Abstract:~3] OR "participatory epidemiology"[Title/Abstract] OR sentinel*[Title/Abstract] OR (("Disease"[Mesh]) AND ("Surveys and Questionnaires"[Mesh] OR "Sentinel Surveillance"[Mesh] OR "Public Health Surveillance"[Mesh] OR "Self Report"[Mesh] OR "Mandatory Reporting"[Mesh] OR "Social Control, Informal"[Mesh] OR "Social Control, Formal"[Mesh] OR "Infection Control"[Mesh] OR "Communicable Disease Control"[Mesh] OR "Behavior Control"[Mesh] OR "Social Control Policies"[Mesh] OR "Data Collection"[Mesh] OR "Censuses"[Mesh] OR "Epidemiological Monitoring"[Mesh] OR "Health Information Exchange"[Mesh] OR "Disclosure"[Mesh] OR "Self Disclosure"[Mesh] OR "Geographic Mapping"[Mesh] OR "Mass Screening"[Mesh] OR "Mandatory Testing"[Mesh])) OR "Disease Notification"[Mesh] |

**Table A1.** *Cont.*

| Key Elements | Components of This Study | Search Terms |
|---|---|---|
| Outcome 2 | Animal disease | "notifiable disease*"[Title/Abstract] OR "reportable disease*"[Title/Abstract] OR "animal disease*"[Title/Abstract] OR "sheep disease*"[Title/Abstract] OR "ovine disease*"[Title/Abstract] OR "goat disease*"[Title/Abstract] OR "caprine disease*"[Title/Abstract] OR "canine disease*"[Title/Abstract] OR "dog disease*"[Title/Abstract] OR "cat disease*"[Title/Abstract] OR "feline disease*"[Title/Abstract] OR "poultry disease*"[Title/Abstract] OR "bird disease*"[Title/Abstract] OR "avian disease*"[Title/Abstract] OR "aquatic disease*"[Title/Abstract] OR "fish disease*"[Title/Abstract] OR "swine disease*"[Title/Abstract] OR "porcine disease*"[Title/Abstract] OR "pig disease*"[Title/Abstract] OR "cattle disease*"[Title/Abstract] OR "bovine disease*"[Title/Abstract] OR "horse disease*"[Title/Abstract] OR "livestock disease*"[Title/Abstract] OR zoonotic[Title/Abstract] OR zoonoses[Title/Abstract] OR Akabane[Title/Abstract] OR "Akabane orthobunyavirus"[Title/Abstract] OR "AKAV"[Title/Abstract] OR "Yaba-7 virus"[Title/Abstract] OR "Tinaroo virus"[Title/Abstract] OR "Sabo virus"[Title/Abstract] OR Anthrax[Title/Abstract] OR "Bacillus anthracis"[Title/Abstract] OR Bluetongue[Title/Abstract] OR "blue tongue"[Title/Abstract] OR "Bovine Tuberculosis"[Title/Abstract] OR "Mycobacterium bovis"[Title/Abstract] OR Brucellosis[Title/Abstract] OR "Brucella abortus"[Title/Abstract] OR "Brucella melitensis"[Title/Abstract] OR "Brucella suis"[Title/Abstract] OR "Gibralter fever"[Title/Abstract] OR "Malta fever"[Title/Abstract] OR "Cyprus fever"[Title/Abstract] OR "undulant fever"[Title/Abstract] OR "Danube fever"[Title/Abstract] OR "Crimean Congo hemorrhagic fever"[Title/Abstract] OR "Khasan virus"[Title/Abstract] OR "Kodzha virus"[Title/Abstract] OR "CCHFV"[Title/Abstract] OR "HAZV"[Title/Abstract] OR "Hazara virus"[Title/Abstract] OR "KHAV"[Title/Abstract] OR "Eastern Equine encephalomyelitis"[Title/Abstract] OR "Epizootic hemorrhagic disease"[Title/Abstract] OR "epizootic haemorrhagic disease"[Title/Abstract] OR "EHD"[Title/Abstract] OR "Foot-and-Mouth Disease"[Title/Abstract] OR "FMD"[Title/Abstract] OR "hoof and mouth"[Title/Abstract] OR Heartwater[Title/Abstract] OR "Ehrlichia ruminantium"[Title/Abstract] OR "Japanese Encephalitis"[Title/Abstract] OR "Japanese B encephalitis"[Title/Abstract] OR Meliodosis[Title/Abstract] OR Melioidosis[Title/Abstract] OR "Burkholderia pseudomallei"[Title/Abstract] OR "Mycobacterium Tuberculosis Complex"[Title/Abstract] OR "M. caprae"[Title/Abstract] OR "Mycobacterium caprae"[Title/Abstract] OR "M. tuberculosis"[Title/Abstract] OR "Mycobacterium tuberculosis"[Title/Abstract] OR Screwworm[Title/Abstract] OR "screw worm"[Title/Abstract] OR "Cochliomyia hominivorax"[Title/Abstract] OR "Chrysomya bezziana"[Title/Abstract] OR Pseudorabies[Title/Abstract] OR "pseudo-rabies"[Title/Abstract] OR "Aujeszky disease"[Title/Abstract] OR "Aujeszky's disease"[Title/Abstract] OR Rabies[Title/Abstract] OR "Rift Valley Fever"[Title/Abstract] OR "Belterra virus"[Title/Abstract] OR "Rift Valley phlebovirus"[Title/Abstract] OR Icoaraci[Title/Abstract] OR Rinderpest[Title/Abstract] OR "RDV"[Title/Abstract] OR "RPV"[Title/Abstract] OR "SARS-CoV-2"[Title/Abstract] OR Surra[Title/Abstract] OR "equine trypanosomosis"[Title/Abstract] OR "Trypanosoma evansi"[Title/Abstract] OR Trichinellosis[Title/Abstract] OR trichinelliasis[Title/Abstract] OR Trichinella[Title/Abstract] OR "Vesicular stomatitis"[Title/Abstract] OR "West Nile virus"[Title/Abstract] OR "Kunjin virus"[Title/Abstract] OR "West Nile flavivirus"[Title/Abstract] OR "WNV"[Title/Abstract] OR "KUNV"[Title/Abstract] OR "Duck viral hepatitis"[Title/Abstract] OR "duck virus hepatitis"[Title/Abstract] OR "Fowl typhoid"[Title/Abstract] OR "Salmonella entirica"[Title/Abstract] OR "Salmonella Gallinarum"[Title/Abstract] OR "avian influenza"[Title/Abstract] OR "bird flu"[Title/Abstract] OR "avian flu"[Title/Abstract] OR "influenza in birds"[Title/Abstract] OR "Highly pathogenic avian influenza"[Title/Abstract] OR "HPAI"[Title/Abstract] OR "Low pathogenic avian influenza"[Title/Abstract] OR "LPAI"[Title/Abstract] OR "Pullorum"[Title/Abstract] OR "Turkey rhinotracheitis"[Title/Abstract] OR "avian metapneumovirus"[Title/Abstract] OR Metapneumovirus[Title/Abstract] OR "TRTV"[Title/Abstract] OR "AMPV"[Title/Abstract] OR "APV"[Title/Abstract] OR "Avian pneumovirus"[Title/Abstract] OR "Newcastle disease"[Title/Abstract] OR "Avian orthoavulavirus"[Title/Abstract] OR "Avian paramyxovirus"[Title/Abstract] OR "NDV"[Title/Abstract] OR "APMV-1"[Title/Abstract] OR "Bovine babesiosis"[Title/Abstract] OR "Babesia bovis"[Title/Abstract] OR "B.bigemina"[Title/Abstract] OR "Babesia bigemina"[Title/Abstract] OR Babesiosis[Title/Abstract] OR "Bovine spongiform encephalopathy"[Title/Abstract] OR "BSE"[Title/Abstract] OR "prion disease*"[Title/Abstract] OR "Contagious bovine pleuropneumonia"[Title/Abstract] OR Pleuropneumonia[Title/Abstract] OR "Mycoplasma mycoides mycoides"[Title/Abstract] OR "Hemorrhagic septicemia"[Title/Abstract] OR "Pasteurella multocida"[Title/Abstract] OR "Lumpy skin disease"[Title/Abstract] OR Theileriosis[Title/Abstract] OR "Theileria annulata"[Title/Abstract] OR "T. annulata"[Title/Abstract] OR "Theileria parva"[Title/Abstract] OR "T.parva"[Title/Abstract] OR Trichomoniasis[Title/Abstract] OR Trypanosomosis[Title/Abstract] OR "Contagious caprine pleuropneumonia"[Title/Abstract] OR Mange[Title/Abstract] OR "Mite Infestation*"[Title/Abstract] OR "sheep scab"[Title/Abstract] OR "Sarcoptes scabiei var ovis"[Title/Abstract] OR "Chorioptes bovis"[Title/Abstract] OR Psoroptes ovis"[Title/Abstract] OR "Psoroptes cuniculi"[Title/Abstract] OR "Psoregates ovis"[Title/Abstract] OR "Nairobi sheep disease"[Title/Abstract] OR "Peste des petitis ruminants"[Title/Abstract] OR Scrapie[Title/Abstract] OR "Sheep pox"[Title/Abstract] OR "goat pox"[Title/Abstract] OR "African horse sickness"[Title/Abstract] OR "Contagious equine metritis"[Title/Abstract] OR "CEM"[Title/Abstract] OR "Taylorella equigenitalis"[Title/Abstract] OR Dourine[Title/Abstract] OR "Trypanasoma equiperdum"[Title/Abstract] OR "Equine infectious anemia"[Title/Abstract] OR "Equine piroplasmosis"[Title/Abstract] OR babesiosis[Title/Abstract] OR "Theileria equi"[Title/Abstract] OR "Babesia caballi"[Title/Abstract] OR "Equine rhinopneumonitis equine herpesvirus-1 myeloencephalopathy"[Title/Abstract] OR "Equine Herpesvirus Myeloencephalopathy"[Title/Abstract] OR Encephalomyelitis[Title/Abstract] OR "EHV1-EHM"[Title/Abstract] OR Glanders[Title/Abstract] OR "Burkholderia mallei"[Title/Abstract] OR Hendra[Title/Abstract] OR "Venezuelan Equine encephalomyelitis"[Title/Abstract] OR VEE[Title/Abstract] OR "Western Equine encephalomyelitis"[Title/Abstract] OR "Chronic wasting disease"[Title/Abstract] OR CWD[Title/Abstract] OR Myxomatosis[Title/Abstract] OR "Rabbit hemorrhagic disease"[Title/Abstract] OR "swine fever"[Title/Abstract] OR "hog cholera"[Title/Abstract] OR "Nipah virus"[Title/Abstract] OR "Swine vesicular disease"[Title/Abstract] OR "Camel pox"[Title/Abstract] OR (dromedary camel*[Title/Abstract] OR (respiratory[Title/Abstract] AND coronavirus[Title/Abstract] OR MERS CoV[Title/Abstract])) OR Leishmaniasis[Title/Abstract] OR exanthema[Title/Abstract] OR "Senecavirus A"[Title/Abstract] OR "Seneca Valley virus"[Title/Abstract] OR Echinococcus Granulosis[Title/Abstract] OR Hydatidosis[Title/Abstract] OR Echinococcosis[Title/Abstract] OR "Echinococcus Multicularis"[Title/Abstract] OR "Johne's disease"[Title/Abstract] OR "Johnes Disease"[Title/Abstract] OR "Mycobacterium avium paratuberculosis"[Title/Abstract] OR "Paratuberculosis"[Title/Abstract] OR "Q Fever"[Title/Abstract] OR "Coxiella burnetii"[Title/Abstract] OR Tularemia[Title/Abstract] OR "Francisella tularensis"[Title/Abstract] OR "Avian chlamydiosis"[Title/Abstract] OR "Chlamydia psittaci"[Title/Abstract] OR Psittacosis[Title/Abstract] OR "Avian infectious bronchitis"[Title/Abstract] OR "Avian infectious laryngotracheitis"[Title/Abstract] OR "Avian mycoplasmosis"[Title/Abstract] OR "Mycoplasma gallisepticum"[Title/Abstract] OR "Mycoplasma synoviae"[Title/Abstract] OR "Infectious bursal disease"[Title/Abstract] OR "Gumboro disease"[Title/Abstract] OR Anaplasmosis[Title/Abstract] OR "Anaplasma marginale"[Title/Abstract] OR "A. marginale"[Title/Abstract] OR "Anaplasma central"[Title/Abstract] OR "Bovine genital campylobacteriosis"[Title/Abstract] OR "Campylobacter fetus venerealis"[Title/Abstract] OR "Bovine viral diarrhea"[Title/Abstract] OR "BVD"[Title/Abstract] OR "mucosal disease"[Title/Abstract] OR "Enzootic bovine leucosis"[Title/Abstract] OR "BLV"[Title/Abstract] OR "Infectious bovine rhinotracheitis"[Title/Abstract] OR "infectious pustular vulvovaginitis"[Title/Abstract] OR "IBR"[Title/Abstract] OR "IPV"[Title/Abstract] OR "IBR/IPV"[Title/Abstract] OR "Malignant catarrhal fever"[Title/Abstract] OR "Malignant Catarrh"[Title/Abstract] OR "Caprine arthritis encephalitis"[Title/Abstract] OR "CAE"[Title/Abstract] OR Encephalitis[Title/Abstract] OR "Contagious agalactia"[Title/Abstract] OR "Mycoplasma agalactiae"[Title/Abstract] OR "Mycoplasma Capricolum capricolum"[Title/Abstract] OR "Mycoplasma putrefaciens"[Title/Abstract] OR "Mycoplasma mycoides mycoides LC"[Title/Abstract] OR "M. Capricolum capricolum"[Title/Abstract] OR "M. putrefaciens"[Title/Abstract] OR "M. mycoides mycoides"[Title/Abstract] OR "M. mycoides mycoides LC"[Title/Abstract] OR "Enzootic abortion of ewes"[Title/Abstract] OR "ovine chlamydiosis"[Title/Abstract] OR Chlamydophila abortus"[Title/Abstract] OR "Maedi-visna/ovine progressive pneumonia"[Title/Abstract] OR "Ovine epididymitis"[Title/Abstract] OR "Brucella ovis infection"[Title/Abstract] OR Epididymitis[Title/Abstract] OR Salmonellosis[Title/Abstract] OR Salmonella[Title/Abstract] OR "Equine influenza"[Title/Abstract] OR "Equine rhinopneumonitis"[Title/Abstract] OR "EHV-1"[Title/Abstract] OR "Herpesvirus 1, Equid"[Title/Abstract] OR "Equine viral arteritis"[Title/Abstract] OR "Equartevirus"[Title/Abstract] OR "EVA"[Title/Abstract] OR "Pigeon fever"[Title/Abstract] OR "Corynebacterium pseudotuberculosis"[Title/Abstract] OR "ulcerative lymphangitis"[Title/Abstract] OR Strangles[Title/Abstract] OR "Streptococcus equi"[Title/Abstract] OR "Porcine Cysticercosis"[Title/Abstract] OR "Taenia solium"[Title/Abstract] OR Cysticercosis[Title/Abstract] OR "Porcine reproductive and respiratory syndrome"[Title/Abstract] OR "PRRS"[Title/Abstract] OR "Transmissible gastroenteritis"[Title/Abstract] OR "TGE"[Title/Abstract] OR "Batrachochytrium dendrobatidis"[Title/Abstract] OR Batrachochytrium[Title/Abstract] OR "Batrachochytrium salamandrivoran"[Title/Abstract] OR Ranavirus[Title/Abstract] OR "Acute hepatopancreatic necrosis disease"[Title/Abstract] OR "Vibrio parahaemolyticus pVA-1 plasmid"[Title/Abstract] OR "Vibrio parahaemolyticus"[Title/Abstract] OR "Crayfish plague"[Title/Abstract] OR "Aphanomyces astaci"[Title/Abstract] OR "Decapod iridescent virus"[Title/Abstract] OR "DIV1"[Title/Abstract] OR "Infectious hypodermal and hematopoietic necrosis"[Title/Abstract] OR "IHHN"[Title/Abstract] OR "Infectious myonecrosis"[Title/Abstract] OR "Necrotizing hepatopancreatitis"[Title/Abstract] OR "Candidatus Hepatobacter penaei"[Title/Abstract] OR "Taura syndrome"[Title/Abstract] OR "White spot disease"[Title/Abstract] OR "white spot syndrome virus"[Title/Abstract] OR "White tail disease"[Title/Abstract] OR "Hypodermic and hematopoietic necrosis baculovirus"[Title/Abstract] OR "White spot baculovirus"[Title/Abstract] OR "WSSV"[Title/Abstract] OR "Chinese baculo-like virus"[Title/Abstract] OR "White spot bacilliform virus"[Title/Abstract] OR "Macrobrachium rosenbergii nodavirus"[Title/Abstract] OR "Yellow head"[Title/Abstract] OR "Roniviridae"[Title/Abstract] OR "Epizotic hematopoietic necrosis disease"[Title/Abstract] OR "Epizootic ulcerative syndrome"[Title/Abstract] OR "EUS"[Title/Abstract] OR "Aphanomyces invadans"[Title/Abstract] OR Gyrodactylosis[Title/Abstract] OR "Gyrodactylus salaris"[Title/Abstract] OR "Infectious haematopoietic necrosis"[Title/Abstract] OR "IHN"[Title/Abstract] OR "Infectious salmon anemia"[Title/Abstract] OR "isavirus"[Title/Abstract] OR "ISA"[Title/Abstract] OR "HPR0"[Title/Abstract] OR "HPR-deleted"[Title/Abstract] OR Alphavirus[Title/Abstract] OR "Red sea bream iridoviral disease"[Title/Abstract] OR "Spring viremia"[Title/Abstract] OR "Spring viremia[Title/Abstract] OR "Tilapia Lake Virus"[Title/Abstract] OR "Viral hemorrhagic septicemia"[Title/Abstract] OR "VHS"[Title/Abstract] OR "abalone herpes virus"[Title/Abstract] OR "Abalone viral ganglioneuritis"[Title/Abstract] OR "Abalone herpesvirus"[Title/Abstract] OR Bonamiosis[Title/Abstract] OR "B. exitiosa"[Title/Abstract] OR "B. ostreae"[Title/Abstract] OR "Bonamia exitosa"[Title/Abstract] OR "Bonamia ostreae"[Title/Abstract] OR Haplosporida[Title/Abstract] OR "Marteilia refringens"[Title/Abstract] OR "Perkinsus olseni"[Title/Abstract] OR "Xenohaliotis californiensis"[Title/Abstract] OR "Koi herpesvirus disease"[Title/Abstract] OR "Perkinsus marinus"[Title/Abstract] OR "Boil disease"[Title/Abstract] OR "Bubble disease"[Title/Abstract] OR "Gill disease"[Title/Abstract] OR "Peduncle disease"[Title/Abstract] OR "Redmouth disease"[Title/Abstract] OR "Yersinia ruckeri"[Title/Abstract] OR "Ulcerative dermal necrosis"[Title/Abstract] OR Vibriosis[Title/Abstract] OR "Whirling disease"[Title/Abstract] OR "Myxobolus cerebralis"[Title/Abstract] OR "Tropilaelaps"[Title/Abstract] OR "Acarapisosis"[Title/Abstract] OR "Acarapis woodi"[Title/Abstract] OR "Paenibacillus larvae"[Title/Abstract] OR "Deformed Wing Virus, Variant C"[Title/Abstract] OR "DWV-C"[Title/Abstract] OR "European foulbrood"[Title/Abstract] OR "Melissococcus plutonius"[Title/Abstract] OR "Slow bee paralysis virus"[Title/Abstract] OR "SBPV"[Title/Abstract] OR "Small hive beetle infestation"[Title/Abstract] OR "Aethina tumida"[Title/Abstract] OR "Varroosis of honey bees"[Title/Abstract] OR Varroa[Title/Abstract] OR Varroidae[Title/Abstract] OR "Sheep Diseases"[Mesh] OR "Goat Diseases"[Mesh] OR "Dog Diseases"[Mesh] OR "Cat Diseases"[Mesh] OR "Poultry Diseases"[Mesh] OR "Bird Diseases"[Mesh] OR "Animal Diseases"[Mesh] OR "Swine Diseases"[Mesh] OR "Cattle Diseases"[Mesh] OR "Horse Diseases"[Mesh] OR "Fish Diseases"[Mesh] OR "Zoonoses"[Mesh] OR "Anthrax"[Mesh] OR "Tuberculosis, Bovine"[Mesh] OR "Brucellosis, Bovine"[Mesh] OR "Hemorrhagic Fever, Crimean-Congo"[Mesh] OR "Hemorrhagic Fever Virus, Crimean"[Mesh] OR "Encephalomyelitis, Eastern Equine"[Mesh] OR "Encephalitis Virus, Eastern Equine"[Mesh] OR "Hemorrhagic Disease Virus, Epizootic"[Mesh] OR "Foot-and-Mouth Disease"[Mesh] OR "Foot-and-Mouth Disease Virus"[Mesh] OR "Heartwater Disease"[Mesh] OR "Ehrlichia ruminantium"[Mesh] OR "Encephalitis, Japanese"[Mesh] OR "Encephalitis Virus, Japanese"[Mesh] OR "Melioidosis"[Mesh] OR "Burkholderia Infections"[Mesh] OR "Mycobacterium tuberculosis"[Mesh] OR "Screw Worm Infection"[Mesh] OR "Pseudorabies"[Mesh] OR "Rabies"[Mesh] OR "Rabies virus"[Mesh] OR "Rift Valley Fever"[Mesh] OR "Rift Valley fever virus"[Mesh] OR "Phlebovirus"[Mesh] OR "Rinderpest virus"[Mesh] OR "Rinderpest"[Mesh] OR "SARS-CoV-2"[Mesh] OR "Trypanosomiasis"[Mesh] OR "Trypanosomiasis, African"[Mesh] OR "Trypanosomiasis, Bovine"[Mesh] OR "Dourine"[Mesh] OR "Trichinellosis"[Mesh] OR "Vesicular stomatitis Indiana virus"[Mesh] OR "Vesicular stomatitis New Jersey virus"[Mesh] OR "Vesicular Stomatitis"[Mesh] OR "West Nile Fever"[Mesh] OR "West Nile virus"[Mesh] OR "Hepatitis, Viral, Animal"[Mesh] OR "Salmonella enterica"[Mesh] OR "Typhoid Fever"[Mesh] OR "Influenza in Birds"[Mesh] OR "Influenza A virus"[Mesh] OR "Metapneumovirus"[Mesh] OR "Newcastle disease virus"[Mesh] OR "Newcastle Disease"[Mesh] OR "Avulavirus"[Mesh] OR "Babesiosis"[Mesh] OR "Encephalopathy, Bovine Spongiform"[Mesh] OR "Prion Diseases"[Mesh] OR "Pleuropneumonia, Contagious"[Mesh] OR "Mycoplasma"[Mesh] OR "Hemorrhagic Septicemia, Viral"[Mesh] OR "Hemorrhagic Septicemia"[Mesh] OR "Pasteurella multocida"[Mesh] OR "Pasteurella Infections"[Mesh] OR "Lumpy Skin Disease"[Mesh] OR "Lumpy skin disease virus"[Mesh] OR "Theileriasis"[Mesh] OR "Trichomonas Infections"[Mesh] OR "Mite Infestations"[Mesh] OR "Scabies"[Mesh] OR "Psoroptidae"[Mesh] OR "Nairobi sheep disease virus"[Mesh] OR "Nairobi Sheep Disease"[Mesh] OR "Peste-des-Petits-Ruminants"[Mesh] OR "Peste-des-petits-ruminants virus"[Mesh] OR "Scrapie"[Mesh] OR "Capripoxvirus"[Mesh] OR "African Horse Sickness"[Mesh] OR "African Horse Sickness Virus"[Mesh] OR "Taylorella equigenitalis"[Mesh] OR "Infectious Anemia Virus, Equine"[Mesh] OR "Equine Infectious Anemia"[Mesh] OR "Herpesvirus 4, Equid"[Mesh] OR "Encephalomyelitis, Equine"[Mesh] OR "Glanders"[Mesh] OR "Hendra Virus"[Mesh] OR "Henipavirus Infections"[Mesh] OR "Encephalomyelitis"[Mesh] OR "Encephalomyelitis, Eastern Equine"[Mesh] OR "Encephalomyelitis, Western Equine"[Mesh] OR "Encephalomyelitis, Venezuelan Equine"[Mesh] OR "Encephalomyelitis, Enzootic Porcine"[Mesh] OR "Encephalomyelitis Virus, Avian"[Mesh] OR "Wasting Disease, Chronic"[Mesh] OR "Myxoma virus"[Mesh] OR "Myxomatosis, Infectious"[Mesh] OR "Hemorrhagic Disease Virus, Rabbit"[Mesh] OR "African Swine Fever"[Mesh] OR "African Swine Fever Virus"[Mesh] OR "Classical Swine Fever Virus"[Mesh] OR "Classical Swine Fever"[Mesh] OR "Nipah Virus"[Mesh] OR "Swine Vesicular Disease"[Mesh] OR "Leishmaniasis"[Mesh] OR "Exanthema"[Mesh] OR "Picornaviridae"[Mesh] OR "Echinococcus"[Mesh] OR "Echinococcus granulosus"[Mesh] OR "Paratuberculosis"[Mesh] OR "Q Fever"[Mesh] OR "Tularemia"[Mesh] OR "Chlamydophila psittaci"[Mesh] OR "Psittacosis"[Mesh] OR "Infectious bronchitis virus"[Mesh] OR "Iltovirus"[Mesh] OR "Mycoplasma Infections"[Mesh] OR "Mycoplasma gallisepticum"[Mesh] OR "Mycoplasma synoviae"[Mesh] OR "Infectious bursal disease virus"[Mesh] OR "Anaplasmosis"[Mesh] OR "Anaplasma marginale"[Mesh] OR "Campylobacter Infections"[Mesh] OR "Bovine Virus Diarrhea-Mucosal Disease"[Mesh] OR "Diarrhea Viruses, Bovine Viral"[Mesh] OR "Diarrhea Virus 1, Bovine Viral"[Mesh] OR "Diarrhea Virus 2, Bovine Viral"[Mesh] OR "Enzootic Bovine Leukosis"[Mesh] OR "Leukemia Virus, Bovine"[Mesh] OR "Infectious Bovine Rhinotracheitis"[Mesh] OR "Herpesvirus 1, Bovine"[Mesh] OR "Malignant Catarrh"[Mesh] OR "Arthritis-Encephalitis Virus, Caprine"[Mesh] OR "Encephalitis"[Mesh] OR "Mycoplasma agalactiae"[Mesh] OR "Chlamydia"[Mesh] OR "Pneumonia, Progressive Interstitial, of Sheep"[Mesh] OR "Brucella ovis"[Mesh] OR "Epididymitis"[Mesh] OR "Salmonella Infections"[Mesh] OR "Salmonella"[Mesh] OR "Influenza A Virus, H3N8 Subtype"[Mesh] OR "Herpesvirus 4, Equid"[Mesh] OR "Herpesvirus 1, Meleagrid"[Mesh] OR "Herpesvirus 1, Equid"[Mesh] OR "Herpesvirus 1, Canid"[Mesh] OR "Herpesvirus 2, Gallid"[Mesh] OR "Herpesvirus 1, Gallid"[Mesh] OR "Equartevirus"[Mesh] OR "Corynebacterium pseudotuberculosis"[Mesh] OR "Streptococcus equi"[Mesh] OR "Cysticercosis"[Mesh] OR "Taenia solium"[Mesh] OR "Porcine Reproductive and Respiratory Syndrome"[Mesh] OR "Porcine respiratory and reproductive syndrome virus"[Mesh] OR "Gastroenteritis, Transmissible, of Swine"[Mesh] OR "Transmissible gastroenteritis virus"[Mesh] OR "Batrachochytrium"[Mesh] OR "Vibrio parahaemolyticus"[Mesh] OR "Aphanomyces"[Mesh] OR "Densovirinae"[Mesh] OR "Taura syndrome virus" [Supplementary Concept] OR "Dicistroviridae"[Mesh] OR "Baculoviridae"[Mesh] OR "Roniviridae"[Mesh] OR "Infectious hematopoietic necrosis virus"[Mesh] OR "Epizootic haematopoietic necrosis virus" [Supplementary Concept] OR "Isavirus"[Mesh] OR "Alphavirus"[Mesh] OR "Alphavirus Infections"[Mesh] OR "Viremia"[Mesh] OR "Hemorrhagic Septicemia, Viral"[Mesh] OR "Hemorrhagic Septicemia"[Mesh] OR "Haplosporida"[Mesh] OR "Amoebic gill disease" [Supplementary Concept] OR "Yersinia ruckeri"[Mesh] OR "Vibrio Infections"[Mesh] OR "Paenibacillus larvae"[Mesh] OR "Deformed wing virus" [Supplementary Concept] OR "Melissococcus plutonius" [Supplementary Concept] OR "Acute bee paralysis virus" [Supplementary Concept] OR "Varroidae"[Mesh] |

## Appendix B

Search strategy for databases in the pilot.

**Table A2.** Web of Science search strategy (performed on 24 January 2022).

| Searches Conducted | Search Terms |
|---|---|
| Search 1 | "disease reporting" (all fields) and behavior (all fields) |
| Search 2 | "disease reporting (all fields) and (socio* or social*) (all fields) |

**Table A3.** Scopus search strategy (performed on 24 January 2022).

| Searches Conducted | Search Terms |
|---|---|
| Search 3 | (TITLE-ABS-KEY (("disease report*" OR surveillance)) AND TITLE-ABS-KEY ((livestock OR cattle OR sheep OR goat OR swine OR pig*)) AND TITLE-ABS-KEY ((socio* OR social O behavior* OR behaviour* OR attitude* OR perception* OR vigil* OR barrier* OR "participatory epidemiology"))) |
| Search 4 | (TITLE-ABS-KEY ((disease AND reporting)) AND TITLE-ABS-KEY ((decision OR "decision process" OR "decision making" OR attitude OR perception)) AND TITLE-ABS-KEY ((detection OR surveillance))) |

## Appendix C

Details of articles included in the pilot scoping review.

**Table A4.** Details of six articles included in the pilot scoping review.

| Article | Country(s) of Study | Population | Animal Disease(s) | Zoonotic Disease(s) | Behavioral Barriers | Behavioral Enablers | Behavioral Interventions |
|---|---|---|---|---|---|---|---|
| Bronner et al. [5] | France | Cattle producers | Bovine brucellosis | Yes | Yes | Yes | No |
| Elbers et al. [3] | Netherlands | Poultry farmers | Avian influenza | Yes | Yes | Yes | No |
| Hamilton-Webb et al. [37] | England | Animal keepers | Exotic livestock diseases | Not specified | No | Yes | No |
| Hopp et al. [38] | Norway | Sheep farmers | Scrapie | No | Yes | Yes | No |
| Tukana et al. [39] | Fiji, PNG, Vanuatu, and the Solomon Islands | Veterinarians | Not specified | Not specified | Yes | No | No |
| Vergne et al. [36] | Bulgaria, Germany, and the Western Part of the Russian Federation | Pig farmers and wild boar hunters | African swine fever | No | Yes | Yes | No |

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
