# Peer review of "Human Dimensions in an Animal Disease Reporting System: A Scoping Review Protocol and Pilot Mapping to Behavioral Frameworks"

_agriculture, doi:10.3390/agriculture14020248_

Round 1

Reviewer 1 Report

Comments and Suggestions for Authors

This paper was appears to have been submitted as a Review article  but what seems to be presented is a proposed approach to undertaking a later review.  If this is incorrect, the authors need to better communicate the intent of the paper and clarify where the results of their scoping review are presented and used. The combination of methods and results made it hard to determine what was done but it seems to me that the authors have proposed a scoping review strategy and then applied it to 6 randomly selected papers from a sub-sample of 125 eligible papers. It was not clear why only a pilot study was used when a relatively small number of eligible papers were found. It was  unclear if the contents of the Discussion were the result of a scoping review, a narrative review or the references were not from any review process. 

The journal webpage states that a review article  should be critical and constructive and offer a comprehensive analysis of the existing literature within a field, identifying current gaps and problems. I do not think this be paper meets that expectation.  If the editor wishes to publish this, it should presented as a project proposal rather than a review. If this were the case, the editor would need to give guidance on the journals review criteria because the current template provided would be inappropriate.

I have opted to recommend rejection because this does not seem to fit the expectations for a review paper and I se no category on the journal's website to publish project proposals or protocols. 

Having said this, I very much encourage the authors to either (1) provide a review article that provides the rationale allowing them to customize a behavioural analysis tool to the animal health sector or wait until they have completed the review and share those results. This is a very important area of scholarship that has received insufficient attention. Their findings will be very informative for future diseases control planning. 

If the authors chose to do so, please pay attention to the generalizability of the extracted literature and how behavioral drivers might work differently in different economic and cultural context. Please also define what you mean by evidence because, as written it seems that you are discounting social science evidence from sources such as interviews, focus groups and other similar sources. Such an definition of evidence would be inappropriate when trying to understand human behaviors. 

Author Response

Thank you for the feedback. We have attached our responses in the attached letter.

Reviewer 2 Report

Comments and Suggestions for Authors

The work presented in this paper sounds interesting and it is based on the aspects of human behaviour that could limit a proper functioning of an animal disease reporting. Animal disease reporting by animal producers and owners is an important component of animal health  surveillance as well as it is a significant aspect of animal welfare monitoring. The manuscript is a review paper that aims to develop a protocol for conducting a scoping review to identify behavioural barriers, enablers, and interventions for animal owners and producers reporting animal diseases to veterinary authorities.

General comments:

The manuscript is not very clear at the beginning (abstract) and at the end (conclusion) about the behavioural frameworks that it deals with. As it is about reporting animal diseases it should be underlined that behavioural frameworks and all given traits (behavioural barriers, enablers,….) are reffering to humans.

Review articles are mainly based on the publications’ survey and in general they include more references than typical research articles. This manuscript includes only 26 references and about 1/3 of them were published 10 years ago or earlier. In my opinion the number of references should be higher and some parts of the manuscript should be more developed (more details are given in specific comments).

The work is generally good, although some minor revisions (in some sections) can be done to improve the quality of the paper and attract a wider readership. Suggested changes are included in the specific comments.

Specific comments:

Abstract: In the first sentence the word ‘human’ could be added before ‘…behavioral barriers’.

The introduction is generally fine but it could be improved using more recent publications that are relevant to the subject.

The section: 2. Methods & Results should be divided into (at least) two sections: 2. Methods and 3. Results or three sections: 2. Methods, 3. Data Extraction and Evaluation, 4. Results.

Conclusion: In the first sentence the word ‘human’ could be added before ‘…behavioral dimensions of animal disease reporting…’.

Only one dot should be placed at the end of the tables' titles.

Author Response

(The authors gave the same response as above.)

Reviewer 3 Report

Comments and Suggestions for Authors

the manuscript describes the protocol devised for the performance of a scoping review on the theme of human behaviors and decisions associated to surveillance systems.

the manuscript is clear and concise, written within the norms of the journal, and adhering to the PRISMA guidelines for both the development of the protocol and its presentation, implementing adequate methodology to deal with each step of the process. However, a few issues must be raised.

on 2.5 and 2.7, you cite that two independent reviewers will perform the classification/critical appraisal, and that discrepancies will be discussed. So the authors intend to deal with all discrepancies by consensus? is there a threshold that will be considered for concordance? if so, what will be used to define it (e.g.: Cohen's Kappa).

Given that the discrepancies between reviewers will be discussed and solved by consensus, how will the authors maintain the process of classification independent between reviewers?

Table 4 (data extracted) is not necessary, and should be presented in text.

given that the manuscript includes a pilot, this should be expressed in the title.

Table 5/6: contents should be justified to facilitate reading.

In the appendix B, there are search strategies for Scopus. However, in the methods, Scopus is not included. Will Scopus be included in the final pool of search databases? and if not, why?

Author Response

(The authors gave the same response as above.)
